# Mechanotransductive Activation of PPAR-γ by Low-Intensity Pulsed Ultrasound Induces Contractile Phenotype in Corpus Spongiosum Smooth Muscle Cells

**DOI:** 10.3390/biomedicines13071701

**Published:** 2025-07-11

**Authors:** Huan Yu, Jianying Li, Zihan Xu, Zhiwei Peng, Min Wu, Yiqing Lv, Fang Chen, Mingming Yu, Yichen Huang

**Affiliations:** 1Department of Urology, Shanghai Children’s Hospital, School of Medicine, Shanghai Jiao Tong University, Shanghai 200062, China; dr_yuhuan@163.com (H.Y.); lijianying@shchildren.com.cn (J.L.); xuzihan@shchildren.com.cn (Z.X.); pengzhiweiys@163.com (Z.P.); wumin@shchildren.com.cn (M.W.); lvyq@shchildren.com.cn (Y.L.); chenfang01@sjtu.edu.cn (F.C.); 2Department of Ultrasound in Medicine, Shanghai Sixth People’s Hospital Affiliated to Shanghai Jiao Tong University School of Medicine, Shanghai 200233, China

**Keywords:** low-intensity pulsed ultrasound, corpus spongiosum, smooth muscle cells, phenotypic transformation, hypospadias

## Abstract

**Background:** Previously, we found that the pathological changes in the corpus spongiosum (CS) in hypospadias were mainly localized within smooth muscle tissue, presenting as a transformation from the contraction phenotype to synthesis. The role of low-intensity pulsed ultrasound (LIPUS) in regulating smooth muscle cells (SMCs) and angiogenesis has been confirmed. **Objectives:** To demonstrate the feasibility of regulating the phenotypic transformation of corpus spongiosum smooth muscle cells (CSSMCs) in hypospadias using LIPUS and to explore the potential mechanisms. **Materials and Methods:** The CSSMCs were extracted from CS in patients with proximal hypospadias. In vitro experiments were conducted to explore the appropriate LIPUS irradiation intensity and duration which could promote the phenotypic transformation of CSSMCs. A total of 71 patients with severe hypospadias were randomly divided into a control group and a LIPUS group to verify the in vivo transition effect of LIPUS. Consequently, the potential mechanisms by which LIPUS regulates the phenotypic transformation of CSSMCs were explored in vitro. **Results:** In vitro experiments showed that LIPUS with an intensity of 100 mW/cm^2^ and a duration of 10 min could significantly increase the expression of contraction markers in CSSMCs and decrease the expression of synthesis markers. Moreover, LIPUS stimulation could alter the phenotype of CSSMCs in patients with proximal hypospadias. RNA sequencing results revealed that peroxisome proliferator-activated receptor gamma (PPAR-γ) significantly increased after LIPUS stimulation. Overexpression of PPAR-γ significantly increased the expression of contraction markers in CSSMCs, and the knockdown of PPAR-γ blocked this effect. **Conclusions:** LIPUS can regulate the transition of CSSMCs from a synthetic to a contractile phenotype in hypospadias. The PPAR-γ-mediated signaling pathway is a possible mechanism involved in this process.

## 1. Introduction

Hypospadias is a common congenital malformation of the male genitourinary system, with an incidence of approximately three cases per thousand male births. Surgical reconstruction of the urethra is currently the primary treatment for hypospadias. However, even with successful surgery during early childhood, about 17% of hypospadias patients still experience a short penis after puberty [1], and approximately 20–25% of patients experience recurrent penile curvature in adulthood [2]. About one-third of patients also suffer from erectile dysfunction in adulthood [3]. These clinical phenomena suggest that developmental abnormalities in hypospadias are not limited to the endoderm-derived urethra alone but may also involve the mesoderm-derived corpus spongiosum (CS) and corpus cavernosum (CC), leading to more complex developmental disorders.

Current research on the development of mesoderm-derived CS and CC in hypospadias is relatively limited. The CS and CC are primarily composed of sinusoid-like structures. Baskin et al. observed irregularly enlarged vascular lacunae in the glans penis of hypospadias fetuses [4]. Kureel et al. [5] used MRI to uncover that the diameter of the CC in hypospadias patients is smaller, and the elasticity of both the CC and the CS is reduced. In our previous studies, we found irregularly enlarged vascular lacunae and thickened vascular smooth muscle layers in the CS of hypospadias patients, with these structural variations positively correlating with the severity of the condition [6]. Our further research demonstrated that the smooth muscle cells (SMCs) in the CS of hypospadias patients transform from a contractile phenotype under physiological conditions to a synthetic phenotype under pathological conditions [7]. In this pathological state, the expression of contractile marker proteins such as α-SMA and SM-22α is downregulated, while cell proliferation, migration, and extracellular matrix synthesis are significantly increased, accompanied by upregulated expression of synthetic marker proteins such as OPN and Collagen I. This phenotypic transformation closely resembles the characteristics of cavernous SMCs in patients with erectile dysfunction [8]. Therefore, reversing the phenotypic transformation of mesoderm-derived CSSMCs holds promise for correcting developmental abnormalities of the corpus spongiosum in hypospadias patients, thereby promoting normal penile development after puberty.

Low-intensity pulsed ultrasound (LIPUS) is an emerging micro-energy medical intervention that modulates cellular functions through its cavitation and mechanical effects. Studies have shown that shear stress can activate the peroxisome proliferator-activated receptor (PPAR) signaling pathway in SMCs, thereby inducing a transformation from a synthetic to a contractile phenotype [9]. PPAR-γ, a nuclear receptor involved in lipid metabolism and anti-inflammatory responses, is also essential in vascular remodeling [10,11]. In ED, the clinical application of LIPUS has gained increasing recognition, as it has been shown to significantly improve the International Index of Erectile Function-Erectile Function score, Erectile Hardness Score, and Self-Esteem And Relationship score [12].

Based on these results, this study focuses on mesoderm-derived CSSMCs in hypospadias, aiming to explore the effects of LIPUS on their phenotypic transformation. Specifically, we investigate whether LIPUS could enhance PPAR-γ activity to promote the transition from the synthetic to the contractile phenotype. This research seeks to provide a theoretical foundation and potential intervention strategies for treating cavernous tissue developmental abnormalities associated with hypospadias.

## 2. Materials and Methods

### 2.1. Patient Enrollment and Sample Collection

This study was approved by the Ethics Committee of Shanghai Children’s Hospital. We recruited 71 children with severe hypospadias, aged from 8.7 months to 4.4 years (mean age: 2 years, median age: 1.66 years), who presented for their first visit between 2023 and 2024. Clinical diagnosis was confirmed by direct clinical examination by pediatric urologists and/or endocrinologists. Patients were excluded if they (1) carried known karyotype abnormalities or abnormalities in the sex-determining region Y (SRY) gene; (2) were diagnosed with hypospadias as a result of disorders of sex development (DSD); (3) showed endocrine disorder; (4) were diagnosed with additional clinical features such as cryptorchidism or micropenis. The 71 children were randomly divided into a control group (n = 40) and an experimental group (n = 31) by simple randomization. Tissue samples were collected from corpus spongiosum surrounding the urethral plate during surgery.

### 2.2. Culture of CSSMCs

The CS specimens were washed with PBS buffer (Servicebio, G4202, Wuhan, China) before being transferred to Petri dishes containing DMEM culture medium (Thermo Fisher Scientific, C11995500BT, Waltham, MA, USA). The tissue was finely cut into pieces and then transferred to a centrifuge tube containing type IV collagenase and separation enzyme(Sigma-Aldrich, C5138, St. Louis, MO, USA). After a 30 min digestion, the single-cell suspensions were collected and seeded into a 24-well cell culture plate after centrifugation and resuspension. The culture medium was carefully placed into an incubator and replaced every two days. The cells were passaged upon reaching 90% confluence. Cells from passages 3–5 were used for further experiments.

### 2.3. Immunofluorescence Staining

Cells were fixed with 4% paraformaldehyde, permeabilized with 0.5% Triton X-100 and then incubated with primary antibodies, α-SMA (Abcam, ab17817, Cambridge, UK, 1:300) and SM-22α(Cell Signaling Technology, 36090, Danvers, MA, USA, 1:200), at room temperature overnight (4 °C). On the following day, cells were washed three times with PBS for 5 min each time and then incubated with secondary antibodies at room temperature for 1 h. The cell nuclei were stained with 4,6-diamino-2-phenylindole (DAPI) for 5 min.

### 2.4. LIPUS Treatment

Primary CSSMCs were cultured in serum-free DMEM for 12 h prior to LIPUS treatment performed using a multi-mode LIPUS therapeutic apparatus (model WBL-BL, F-probe) from Beijing Wanboli Medical Instrument Co., Ltd. (Beijing, China). Then, cells were placed on the probe (with a frequency of 1.7 MHz, a pulse duration of 200 μs and a duty cycle of 20%) and treated with an intensity of 25 mW/cm^2^, 100 mW/cm^2^, or 500 mW/cm^2^ to determine the appropriate intensity. After LIPUS treatment, the culture medium was replaced with complete culture medium, and cells were collected 24 h later.

LIPUS treatment was also performed on children with hypospadias to explore the effects in vivo. The patients in the LIPUS group received LIPUS treatment (100 mW/cm^2^, 10 min stimulation) 24 h before Byars stage I surgery, and the patients in the control group received the same treatment but with the LIPUS device turned off. During surgery, CS tissues were harvested and separately preserved in liquid nitrogen or 4% paraformaldehyde (Servicebio, G1101, Wuhan, China) for different analyses.

### 2.5. Western Blot

Cells or tissue samples were lysed in RIPA buffer using a low-temperature homogenizer. After centrifugation, protein concentration was measured using the BCA method. Equal amounts of protein were separated by SDS-PAGE and transferred onto a PVDF membrane, which was blocked for 1 h at 25 °C. Membranes were incubated overnight at 4 °C with primary antibodies α-SMA (Abcam, ab17817, Cambridge, UK, 1:1000), SM-22α (CST, 36090, Danvers, MA, USA, 1:1000), OPN (CST, 88742, Danvers, MA, USA, 1:1000), Collagen I (CST, 72026, Danvers, MA, USA, 1:1000), and GAPDH (CST, 2118, Danvers, MA, USA, 1:3000). After washing, membranes were incubated with secondary antibodies at room temperature for 2 h, and protein bands were detected using ECL chemiluminescence.

### 2.6. Real-Time Quantitative PCR (qRT-PCR)

Total RNA from the CS tissue was extracted using Trizol reagent (Invitrogen, T9424, Waltham, MA, USA, USA), and reverse-transcribed into cDNA using a reverse transcription kit (Takara, RR036A, Shiga, Japan). PCR amplification was performed in a 10 µL reaction mixture, consisting of 5 µL of 2× SYBR Green PCR Master Mix(Yeasen Biotechnology, 11188ES08, Shanghai, China), 1 µL of forward and reverse primers, 1 µL of cDNA, and 2 µL of water. The PCR conditions were as follows: pre-denaturation at 95 °C for 10 min, followed by 40 cycles of denaturation at 95 °C for 10 s, annealing at 60 °C for 30 s, and extension at 72 °C for 30 s. Relative gene expression was calculated using the 2-ΔCT method. Primer sequences are provided in the Appendix A.

### 2.7. Hematoxylin and Eosin (HE) Staining

CS tissue specimens were fixed in 4% paraformaldehyde, embedded in paraffin, and sectioned into 5 µm-thick slices. The sections were deparaffinized, stained with hematoxylin and eosin, and examined under a microscope.

### 2.8. mRNA Differential Expression

CSSMCs were preserved in RNA and later sent to LC-Bio Technology CO (Hangzhou, China) for sequencing. Total RNA was extracted using the TRIzol reagent according to the manufacturer’s protocol. RNA quantity and purity were analyzed using a Bioanalyzer 2100 and an RNA 6000 Nano LabChip Kit (Agilent, 5067-1511, Santa Clara, CA, USA,). High-quality RNA samples with RIN numbers > 7.0 were used to construct a sequencing library. Sequencing was carried out on an Illumina Novaseq™ 6000 (Illumina, San Diego, CA, USA).

For experiments with duplicate samples, differential gene expression was analyzed with the DESeq2 package in R according to the package manual (https://bioconductor.org/packages/release/bioc/html/DESeq2.html, last accessed 2 July 2025). In the absence of duplicate samples, DESeq2 was still used to identify differentially expressed genes between the sample groups, applying the criteria of |log2FC| ≥ 1 and *p*-value ≤ 0.05. Differentially expressed mRNAs from both the control and LIPUS groups were visualized in a scatter plot and analyzed for significant expression differences.

### 2.9. Cell Transfection

Lentiviral vectors were constructed by Shanghai Jikai Gene Medical Technology Co., Ltd. (Shanghai, China) and included PPAR-γ overexpression RNA (oeRNA) and PPAR-γ siRNA. Lentiviral particles overexpressing PPAR-γ, blank lentiviral particles, and lentiviral particles containing PPAR-γ siRNA were co-transfected into 293T cells using packaging and envelope plasmids in RPMI 1640 medium containing 10% fetal bovine serum to produce lentiviral particles. Primary CSSMCs were subsequently transfected with lentiviral particles (1 × 10^8^ TU/mL).

### 2.10. Statistical Analysis

Statistical analysis was performed using SPSS version 24.0 and GraphPad Prism 9.0 software. Data are presented as mean ± standard deviation. Normality and homogeneity of variance were assessed before statistical testing. For normally distributed data with equal variances, a *t*-test was used for two-group comparisons, and one-way ANOVA was used for comparisons among three or more groups. For non-normally distributed data or unequal variances, the Mann–Whitney test was applied for two-group comparisons, and the Kruskal–Wallis test was used for multiple-group comparisons. A *p*-value < 0.05 was considered statistically significant.

## 3. Results

### 3.1. Identification of Primary CSSMCs

The CSSMCs exhibited an elongated spindle shape under a light microscope (Figure 1A) and were both SM-22α and α-SMA positive according to immunofluorescence (Figure 1B–E).

### 3.2. In Vitro Phenotypic Transformation of CS-SMCs Stimulated by LIPUS

In order to determine the optimal intensity and duration of LIPUS, we used different ultrasound intensities (25 mW/cm^2^, 100 mW/cm^2^, 500 mW/cm^2^) to treat CSSMCs for different durations (5 min, 10 min, 15 min). The qRT-PCR analysis revealed the intensity and duration-dependent phenotypic modulation in CSSMCs following LIPUS stimulation (Figure 2). When the ultrasound stimulation duration was 5 min, none of the three intensity levels showed significant changes in contractile-type RNA markers of the cells (Figure 2A). However, following 10 min of ultrasound exposure, CSSMCs treated with 100 mW/cm^2^ and 500 mW/cm^2^ exhibited a significant upregulation of contractile markers (α-SMA, SM-22α) and downregulation of synthetic markers (Collagen I, OPN) (Figure 2B). After 15 min of stimulation, only the 100 mW/cm^2^ group showed significant phenotypic transformation (Figure 2C). These results suggest that 10 min stimulation at 100 mW/cm^2^ or 500 mW/cm^2^ most effectively promotes the phenotypic transition of CSSMCs from a synthetic to a contractile state.

To further assess the protein-level changes in phenotypic markers, Western blot analysis was performed on CSSMCs stimulated with ultrasound intensities of 25 mW/cm^2^, 100 mW/cm^2^, and 500 mW/cm^2^ for 10 min. Results showed that stimulation at 100 mW/cm^2^ for 10 min significantly increased the expression of α-SMA and SM-22α, while markedly decreasing Collagen I and OPN levels. These findings confirm that LIPUS at 100 mW/cm^2^ intensity for 10 min effectively promotes the phenotypic transformation of CSSMCs.

To determine whether LIPUS could induce apoptosis in CSSMCs, we performed a cell viability test. Flow cytometry results showed that 25 mW/cm^2^ and 100 mW/cm^2^ ultrasound intensities did not induce apoptosis in CSSMCs. However, stimulation of CSSMCs at an ultrasound intensity of 500 mW/cm^2^ produced a pro-apoptotic effect (Figure 2I–J). Therefore, LIPUS with an intensity of 100 mW/cm^2^ and a duration of 10 min was used in the subsequent experiments.

### 3.3. In Vitvo Validation of LIPUS’S Phenotypic Transformation Effect

We investigated the effect of LIPUS stimulation (preoperative stimulation for 10 min at an intensity of 100 mW/cm^2^) on cavernous remodeling in children with hypospadias. The CS tissues were obtained from children with hypospadias during surgery for subsequent studies. The results of qRT-PCR and Western blot analysis showed that α-SMA and SM-22α were increased in both mRNA and protein levels in the LIPUS treatment group, while Collagen I and OPN levels were decreased (Figure 3A–I). Histological analysis further confirmed these results (Figure 3J), indicating that LIPUS stimulation could alter the phenotype of CS smooth muscle in hypospadias in vivo.

### 3.4. Mechanism of CSSMCs Phenotypic Transformation by LIPUS

CSSMCs from hypospadias patients treated with or without LIPUS (100 mW/cm^2^, 10 min) were subjected to transcriptome sequencing. By comparing the gene expression of the LIPUS group with the control group (absolute log2FC ≥ 1, *q*-value < 0.05), a total of 424 differentially expressed genes were found, including 162 upregulated genes and 262 downregulated genes in the LIPUS group (Figure 4A). This indicates that the transcriptome of CSSMCs was changed following LIPUS stimulation. KEGG functional enrichment analysis of differentially expressed genes in CSSMCs in the control group and the LIPUS group revealed a pathway related to phenotypic transformation—the PPAR signaling pathway (Figure 4B). The protein and mRNA expression levels of PPAR-γ in CSSMCs from the control group and from the LIPUS group were detected. The results show that compared with the control group, the protein and mRNA expression levels of PPAR-γ in the LIPUS group were increased (*p* < 0.01). (Figure 4C–E) This indicates that stimulation with LIPUS increased the expression level of PPAR-γ in the CSSMCs of hypospadias patients.

### 3.5. PPAR-γ’s Involvement in the Biological Process Phenotypic Transformation of CSSMCs

PPAR-γ is expressed in vascular smooth muscle cells (VSMCs) and plays a role in the vascular system by directly or indirectly regulating genes participating in the regulation of cell proliferation, migration and vascular function [13]. To examine the role of PPAR-γ in the LIPUS-mediated phenotypic modulation of CSSMCs, we used lentivirus to alter the expression of PPAR-γ in CSSMCs. Following PPAR-γ overexpression, the expression of contractile markers in the CSSMCs of hypospadias patients increased, and the expression of synthetic markers decreased. Western blot analysis showed that the protein expression levels of α-SMA and SM-22α in the oe-PPAR-γ group were significantly higher than those in NC group. The protein expression levels of Collagen I and OPN in the oe-PPAR-γ group also significantly decreased (Figure 5A–E). After PPAR-γ interference, the expression of contractile markers in CSSMCs in hypospadias decreased, while the synthetic markers increased. Western blot results showed that the protein expression levels of α-SMA and SM-22α in the si-PPAR-γ group were lower than those in the NC group (*p* < 0.05). The protein expression levels of Collagen I and OPN in the si-PPAR-γ group were increased (*p* < 0.05). PPAR-γ interference blocked the regulation of LIPUS-stimulated phenotypic transformation of CSSMCs in hypospadias (*p* < 0.01) (Figure 5F,G).

## 4. Discussion

In this study, we investigated the phenotypic transformation of CSSMCs derived from hypospadias patients under LIPUS stimulation. Our results demonstrated that LIPUS promoted the phenotypic transition from a synthetic to a contractile phenotype in CSSMCs by upregulating the expression of the peroxisome proliferator PPAR-γ. To our knowledge, this is the first report to explore LIPUS as a biophysical modality for regulating pathological smooth muscle phenotypes in hypospadias.

SMC phenotypic switching is a bidirectional and highly plastic process, which is tightly regulated by environmental stimuli. Under most physiological conditions, SMCs exhibit a contractile phenotype but can undergo phenotypic transitions between contractile and synthetic phenotype [14]. For example, VSMCs transform from a contractile phenotype in response to differentiation signals such as platelet-derived growth factor (PDGF-BB), which maintains the structure and function of mature blood vessels [15]. Conversely, when vascular injury or cytokine stimulation occurs, Kruppel-like transcription factor 4 (KLF4) induces the transformation of VSMCs to the synthetic type for vascular wall repair and regeneration [16]. Epigenetic mechanisms, such as DNA methylation (suppressing SMα-actin expression) and miR-143/145 (inhibiting KLF4), further modulate VSMC plasticity [17,18]. Importantly, a similar dysregulation of smooth muscle phenotypes contributes to erectile dysfunction through increased reactive oxygen species (ROS) and diminished nitric oxide signaling [19,20].

Intriguingly, our previous and current findings suggest that CSSMCs also exhibit a similar capacity for phenotypic modulation. The CS is a sinusoid-like erectile structure composed of a dense vascular network, where vascular endothelial cells form the luminal surface and are circumferentially enveloped by SMCs. This structural composition shares remarkable similarities with the architecture of systemic blood vessels, particularly in the dynamic behavior of their resident SMCs [21]. In hypospadias, these cells predominantly present as synthetic-like phenotypes, which disrupt the structural integrity of the corpus spongiosum and may contribute to urethral instability and surgical failure.

Compared to the fruitful achievements in VSMC research, investigations into the phenotypic transformation of CSSMCs in hypospadias remain in their infancy. Our study reveals that the transition of CSSMCs mirrors the well-described VSMC switching process but may involve unique regulatory mechanisms due to the specialized function of the corpus spongiosum in erectile physiology. Unlike arterial vessels where the primary role of VSMCs is vascular tone regulation [21], CSSMCs coordinate dynamic changes in penile turgor, requiring tight modulation of contractile and relaxation states [22]. This functional distinction suggests that while CSSMC phenotypic plasticity is mechanistically related to that of VSMCs, it is likely governed by a distinct molecular environment.

LIPUS has emerged as a versatile tool for therapy, leveraging mechanical and cavitation effects to modulate cellular behavior. Its safety profile has enabled clinical applications in urology, including for chronic prostatitis/chronic pelvic pain syndrome [23], erectile dysfunction [24], and stress urinary incontinence [25]. Our findings show that LIPUS reverses the synthetic phenotype of CSSMCs both in vitro and in vivo, without inducing apoptosis. Transcriptomic sequencing identified significant upregulation of the PPAR-γ signaling pathway following LIPUS exposure. Activation of PPAR-γ has been shown to inhibit VSMC proliferation and neointimal hyperplasia through the modulation of gene expression and promotion of nitric oxide bioavailability via eNOS activation [26,27]. In our model, PPAR-γ overexpression increased expression of α-SMA and SM-22α while suppressing that of OPN and Collagen I. Conversely, knockdown of PPAR-γ abrogated the LIPUS-induced phenotypic modulation, confirming its central role. Notably, emerging research demonstrates that LIPUS exposure effectively blocks VSMC pathological transformation through dual mechanisms: suppression of miR-17-5p and the potentiation of PPAR-γ signaling [28].

The emerging literature further supports the involvement of PPAR-γ in mechanotransduction. Studies show that LIPUS enhances PPAR-γ-mediated adiponectin expression, improving insulin sensitivity and reducing inflammatory cytokine production in macrophages [29,30]. In bone and cartilage, PPAR-γ is implicated in LIPUS-induced osteogenesis and chondrogenesis [31]. Recent evidence also highlights that LIPUS may also improve vascular function by enhancing eNOS activity, an effect closely associated with the activation of PPAR [32].

The current research endeavors still present certain limitations. Firstly, the in vivo study outcomes indicate that the histological alterations of the corpus spongiosum subjected to short-term LIPUS stimulation are not remarkable, which might be attributed to the insufficient number of irradiations. Secondly, directly applying the LIPUS stimulation intensity and duration obtained from in vitro experiments to human studies might not result in achieving the optimum therapeutic effects due to the barrier presented by human skin. Hence, the optimal treatment parameters for the application of LIPUS stimulation in humans still require further exploration. Additionally, the regulatory mechanism through which LIPUS modifies the expression of PPAR-γ remains unknown at present.

## 5. Conclusions

Low-intensity pulsed ultrasound can regulate the transformation of CSSMCs from the synthetic to the contractile type in hypospadias. The PPAR-γ-mediated signaling pathway may be a possible mechanism by which LIPUS regulates the phenotypic transformation of CSSMCS in hypospadias. Treatment based on low-intensity pulsed ultrasound may be a promising adjuvant treatment for hypospadias patients undergoing surgery.

## Figures and Tables

**Figure 1 biomedicines-13-01701-f001:**
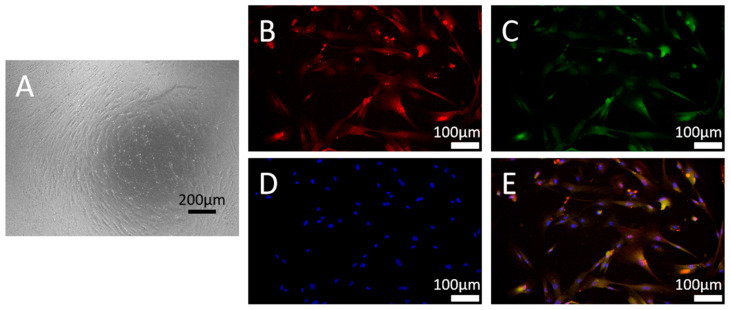
Identification of primary corpus spongiosum smooth muscle cells (CSSMCs). Representative immunofluorescence images of CSSMCs under a light microscope (**A**), labeled with α-SMA (**B**), SM-22α (**C**), DPI (**D**), and merged image (**E**).

**Figure 2 biomedicines-13-01701-f002:**
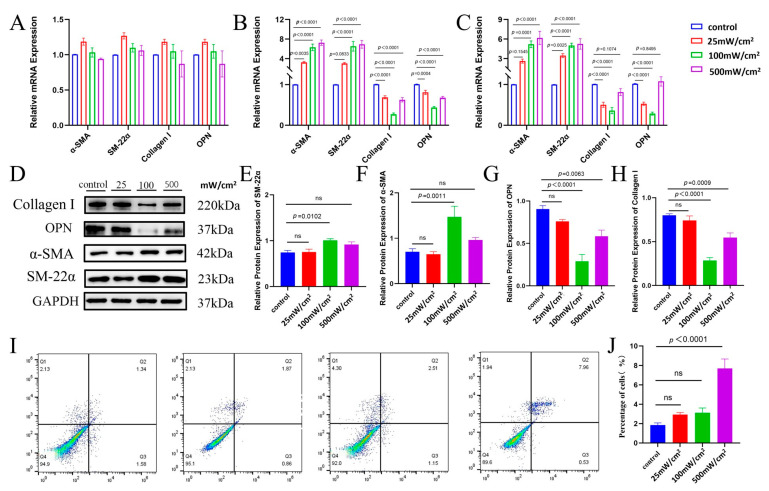
The results of CSSMCs phenotypic transformation stimulated by different ultrasound intensities and durations. (**A**–**C**) qRT-PCR detection of mRNA levels of CSSMCs phenotypic transformation markers under different intensities with 5 min (**A**), 10 min (**B**), and 15 min (**C**) of ultrasound stimulation, respectively (n = 10). (**D**–**H**) Protein expression levels of Collagen I, OPN, α-SMA and SM-22α as determined by Western blot in CSSMCs stimulated for 10 min by LIPUS (n = 3). ns: non-significant. (**I**,**J**) The results of flow cytometry on apoptosis in CSSMC stimulated by different ultrasound intensities (n = 3).

**Figure 3 biomedicines-13-01701-f003:**
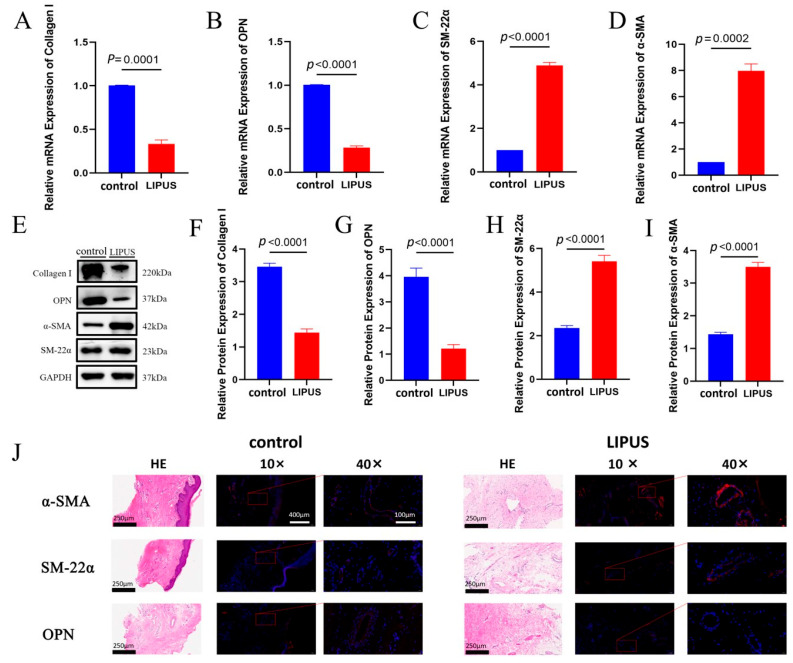
Effect of LIPUS stimulation on phenotypic transformation in the corpus spongiosum of children with hypospadias. (**A**–**D**) mRNA levels of Collagen I, OPN, α-SMA and SM-22α as determined by qRT-PCR in corpus spongiosum irradiated by LIPUS (n = 10). (**E**–**I**) Protein expression levels of Collagen I, OPN, α-SMA and SM-22α as determined by Western blot in corpus spongiosum irradiated by LIPUS (n = 3). (**J**) Histological evaluation including HE, OPN, α-SMA and SM-22α staining. The scale bar represents 250 µm in HE staining image.

**Figure 4 biomedicines-13-01701-f004:**
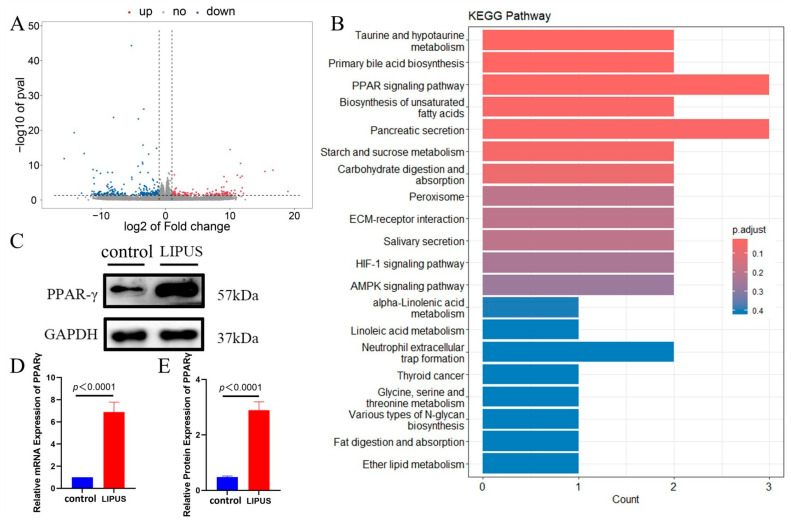
Transcriptome sequencing revealed the role of LIPUS in the phenotypic transformation of CSSMCs. (**A**) Volcano plots of differentially expressed genes. The *x*-axis corresponds to log2 (Fold Change) and the *y*-axis corresponds to −log10 (*p*-adjusted value) (n = 3). (**B**) KEGG enrichment analysis of CSSMC differentially expressed genes in the control and LIPUS groups. (**C**–**E**) Expression levels of PPAR-γ in the CSSMCs of the control and LIPUS groups (n = 3).

**Figure 5 biomedicines-13-01701-f005:**
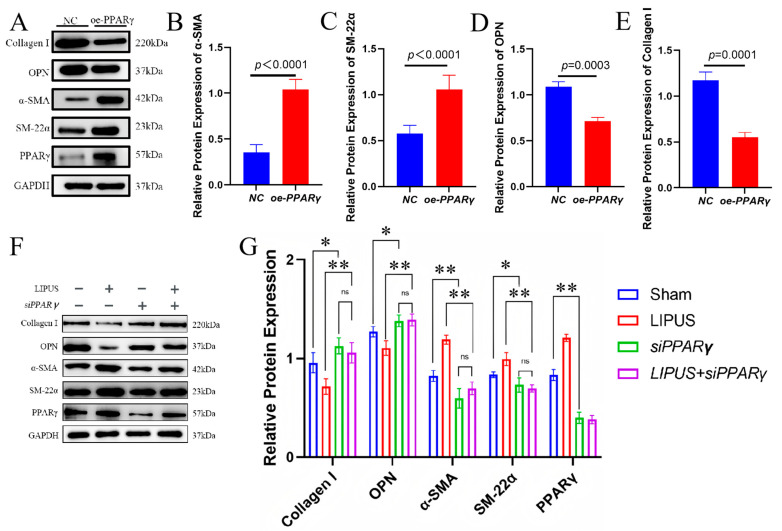
PPAR-γ was involved in the regulation of phenotypic transformation of CSSMCs. (**A**–**E**) Protein expression levels of Collagen I, OPN, α-SMA, SM-22α, and PPAR-γ in CSSMC following overexpression of PPAR-γ determined by Western blot analysis (n = 3). (**F**–**G**) Protein expression levels of Collagen I, OPN, α-SMA, SM-22α, and PPAR-γ in CSSMCs from different groups determined by Western blot analysis. * *p* < 0.05, ** *p* < 0.01 (n = 3), ns: non-significant.

## Data Availability

All data generated or analyzed during this study are included in this published article. The raw sequencing data are available from the corresponding author upon reasonable request.

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
