# Peer review of "Mechanotransductive Activation of PPAR-γ by Low-Intensity Pulsed Ultrasound Induces Contractile Phenotype in Corpus Spongiosum Smooth Muscle Cells"

_biomedicines, 2025, doi:10.3390/biomedicines13071701_

Round 1
Reviewer 1 Report
Comments and Suggestions for Authors
The present study explore the potential applications of LIPUS in patients with hypospadias,attending surgical repair,in order to improve the long term postoperatory results.
Comments:
-row 13 this is the conclusion,not a background
-row 15 "from contraction phenotype to synthetis" did you meaned synthesis?
-row 29 "PPAR gamma" Please provide full name when first utilised in text.
-row 49-51 you discuss about cavernous tissue,but your study is about corpus spongiosum.
-The introduction part explain well the rationale for this study,what is known about LIPUS and hypospadias,and the possible action mechanism of LIPUS in hypospadias patients undergoing surgery.
-Method. Please specify more clear,the prospective nature of the study and the time period.
-row 113 "effcets" please correct.
-row 114 "received the same treatment" Please detail what surgical treatment did you applied.
-row 308-310 This sentence,about PPAR gamma should be inserted in the introduction part.
-row 340 "for hypospadias patients with hypospadias" Please replace "with hypospadias" with undergoing surgery.
-
Author Response
1.row 13 this is the conclusion, not a background
Response: This finding stems from our earlier study and serves as foundation of this study. We have emphasized in the background.
2.row 15 "from contraction phenotype to synthetis" did you mean synthesis?
Response: We have amended “from contraction phenotype to synthetis” in line 15 to “from contraction phenotype to synthesis”.
3.row 29 "PPAR gamma" Please provide full name when first utilised in text.
Response: We have revised the manuscript by providing the full name “peroxisome proliferator-activated receptor gamma (PPARγ)” at its first mention in the manuscript, with the abbreviated form (PPARγ) used thereafter, as per standard conventions.
4.row 49-51 you discuss about cavernous tissue, but your study is about corpus spongiosum.
Response: Thank you for your careful review. The developmental abnormalities in hypospadias involve both mesoderm-derived corpus spongiosum (CS) and corpus cavernosum (CC). Due to ethical considerations, only the CS tissue was involved in this study.
5.The introduction part explain well the rationale for this study, what is known about LIPUS and hypospadias, and the possible action mechanism of LIPUS in hypospadias patients undergoing surgery.
Response: We sincerely appreciate your feedback on our introduction section.
6.Method. Please specify more clear, the prospective nature of the study and the time period.
Response: We have provided a more detailed description of the Patient Enrollment and Sample Collection section in the Methods.
7.row 113 "effcets" please correct.
Response: Thank you for pointing this out. We have amended “effcets” in row 113 to “effects”.
8.row 114 "received the same treatment" Please detail what surgical treatment did you applied.
Response: These patients received Byars stage I surgery, which we have added to the methods.
9.row 308-310 This sentence, about PPAR gamma should be inserted in the introduction part.
Response: We have inserted this into the introduction part.
10.row 340 "for hypospadias patients with hypospadias" Please replace "with hypospadias" with undergoing surgery.
Response: We appreciate this helpful suggestion. We have revised the text to read: "for hypospadias patients undergoing surgery" to improve clarity and precision.
Reviewer 2 Report
Comments and Suggestions for Authors
In this article “Mechanotransductive Activation of PPAR-γ by Low-Intensity Pulsed Ultrasound Induces Contractile Phenotype in Corpus Spongiosum Smooth Muscle Cells” authors investigate the molecular effects of LIPUS over the CSSMCs in vitro with cell cultures and tissue from hypospadias patients. First they discover a transition from synthetic to contractile phenotype of CSSMCS. They identified two related proteins with contractile phenotype, a-SMA and SM-22-a, which showed an increase after LIPUS treatment in cells and tissues. In addition, two proteins related to the synthetic phenotype, CollagenI and OPN, both show a decrease after LIPUS treatment. RNAseq between control and LIPUS treatment show changes in expression genes, and detect a PPARg as an important factor that might control such phenotype change from synthetic to contractile. PPARg have increased after LIPUS treatment. They verify PPAR g regulation through overexpression or silencing. They found that PPAR g regulated the same analyzed proteins that LIPUS treatment, since PPAR g overexpression increased aSMA and SM-22-a and diminished CollagenI and OPN. On the other hand the PPAR g silencing leads to opposite results, however LIPUS treatment was unable to revert due PPAR g being involved in such process. In addition, apoptosis was negligible in LIPUS at 100mW/cm2, where effects were observed,
This article is very interesting and has good uses as a therapeutic treatment. It is interesting to know how ultrasound can change gene expression for differentiation of cells. This effort is well directed to the hypospadia problem in children.
I have some concerns about the work:
1) Author mentions the number of participants in protocolos, but in experiments it is unknown the number of samples analyzed for each experiment, I suggest putting the N for each experiment.
2) The study was approved by the Ethics committee, however the authors do not mention informed consent and assentiment signed by parents and children. This requirement must be met in this vulnerable group of patients.
3) Although RNAseq only analyzed control vs. LIPUS was only one experiment, it would be nice to know other details about the depth of sequencing to know the validity of RNA sequencing.
Minor concerns
1) in Figure 2 , D panel , says CollgenI instead of a CollagenI
2) in Line 113 to correct effcets .
Author Response
1.Author mentions the number of participants in protocolos, but in experiments it is unknown the number of samples analyzed for each experiment, l suggest putting the N for each experiment.
Response: We sincerely appreciate this valuable suggestion. We have now systematically added the exact sample size (N) for each experimental analysis in the Results section.
2.The study was approved by the Ethics committee, however the authors do not mention informed consent and assentiment signed by parents and children. This requirement must be met in this vulnerable group of patients.
Response: We sincerely appreciate the reviewer’s valuable comment regarding the informed consent and assent are, especially when involving vulnerable populations such as children. Written informed consent was obtained from all parents or legal guardians in our study, and the informed consent used in this study has been attached, and signed informed consent forms are also available under reasonable request.
3.Although RNAseq only analyzed control vs. LIPUS was only one experiment, it would be nice to know other details about the depth of sequencing to know the validity of RNA sequencing.
Response: We agree that providing additional details about the sequencing depth and quality would enhance the transparency and validity of our results. Below, we have included the requested information, which has also been added to the Methods section.
4.in Figure 2D panel , says Collgen I instead of a Collagen I
Response: We have corrected the labeling errors in Figure 2D and double-checked the labeling of the Figures in the article to ensure their accuracy.
5.in Line 113 to correct effects.
Response: We have amended “effcets” in row 113 to “effects”.